# *Hemigraphis alternata* Leaf Extract Incorporated Agar/Pectin-Based Bio-Engineered Wound Dressing Materials for Effective Skin Cancer Wound Care Therapy

**DOI:** 10.3390/polym15010115

**Published:** 2022-12-27

**Authors:** Jijo Koshy, Dhanaraj Sangeetha

**Affiliations:** Department of Chemistry, SAS, Vellore Institute of Technology, Vellore 632 014, Tamil Nadu, India

**Keywords:** wound therapy, biopolymers, SKMEL-26 cancer cells, solution casting, agar/pectin, *Hemigraphis alternata*

## Abstract

The rapidly expanding area of regenerative medicine may soon enter a new phase owing to developments in biomaterials and their application in generating new tissues. Chemicals and synthetic drugs are currently the subject of heated debate due to their effects on human health and the environment. Therefore, scientists seek out new products and procedures that are harmless to both the environment and human health concerns. Bio-based materials provide excellent functional qualities with a variety of applications. This study resulted in the development of a film with antimicrobial, hydrophilic, and anti-cancer properties, which is most beneficial in the medical sectors. In this study, we developed a blended biodegradable film containing agar and pectin (AP), with excellent surface functional properties framed through a casting technique. Additionally, the property can be changed by the addition of extract of *hemigraphis alternata* (HA) extract. The incorporation of extract in AP (APH) can be used for anti-cancer wound care therapy. The fabricated film is biodegradable, biocompatible, and non-toxic. This material is entirely based on a green methodology, and it was prepared in a concise manner without the use of any hazardous solvents. Based on the overall nature of biopolymer, the prepared material is a promising alternative to our society.

## 1. Introduction

The majority of recent studies currently focused on transforming sustainable materials into intelligent ones attempt to increase the materials’ necessity. Herbal-derived phytocompounds are the generally used traditional treatment of skin lesions and also the usage of curing wounds derived from eco-friendly biodegradable materials are becoming more popular nowadays. Wound healing is one of the most challenging clinical problems, which has increased the demand for wound management. In the majority of situations, wound dressings must be put to the wounded region to avoid infection and to act as a temporary skin substitute layer [1]. Antimicrobial resistance has severely reduced the therapeutic options for treating skin infections. As a result, treating non-healing chronic wounds is a big issue for health services all over the world, causing significant socioeconomic harm to those who are affected [2]. The ideal wound dressing should maintain a moist environment in the wound area, absorb exudates from the wound surface, and act as a barrier against bacteria while also allowing for customized gaseous exchange [3]. Additionally, the ideal wound dressing should be non-toxic, non-allergenic, and possess excellent cytocompatibility and antibacterial qualities in order to expedite the wound healing process.

The term “cancer” is used to refer to a group of more than 200 diseases that are characterized as defects in the control of tissue growth brought on by mutations in the genes that control cell proliferation and differentiation. Melanoma, which arises from cutaneous, mucosal, and uveal melanocytes, is the most aggressive form of skin cancer. Melanocytes are the origin of the most aggressive form of skin cancer, leading to cutaneous melanoma [4]. Melanoma accounts for only 1% of all malignant skin cancers, in comparison to other skin injuries. Despite recent breakthroughs in therapeutic techniques, melanoma remains the most dangerous form of skin cancer, with a five-year survival rate of only 15 to 20% [5]. SK-MEL-26 is one of a number of melanoma cell lines derived from tumor samples obtained from patients. According to the article’s review, the plant extracts exhibited high anticancer and antiproliferative activity. Medicinal herbs have been found to provide a number of biological benefits against a variety of ailments, including cancer, microbial infections, and urolithiasis, as well as other disorders connected to oxidative stress [6,7,8,9,10]. Khan et al. reported that the discovery of the alkaloid in Catharanthus roseus (Apocynaceae) ushered in a new era of anticancer agents derived from plants [11]. In both in vitro and in vivo experiments, these substances disrupt numerous aspects of tumor development and growth. They cause cell cycle arrest in G1 or G2/M phase and stop the growth of different tumor cell lines by inducing apoptosis [12].

Polysaccharides, proteins, and lipids are the most common biopolymers used in the formation of edible films. Polysaccharides, for instance, are inexpensive and have good functional qualities, therefore they may be utilized as film-forming materials. Through hydrogen bonding, polysaccharide molecules may connect and form a continuous network [4]. Agar, which is commercially extracted from seaweeds, is one of the most typical polysaccharide basic materials that has been thoroughly investigated [13,14,15]. Pectin, on the other hand, is a significant structural component of cell walls that is mostly composed of partly methyl esterified poly (α-d-galacturonic acid) with rhamnose inserts in the backbone and neutral sugar side chains. It is readily biodegradable and water soluble [16]. When combined with natural bioactive substances, edible films containing pectin have antibacterial effects against foodborne pathogens [17]. Blending pectin and agar to generate a hybrid composite improves the physical characteristics of the film while overcoming the limitations of each polymer [18].

Another bioactive functional compound is *Hemigraphis alternata*, (Blume), a member of the Acanthaceae family, regarded as a promising plant by tribal healers in southern India. The plant’s leaves are used in folk medicine for a variety of treatments, including treating fresh wounds, cuts, ulcers, and inflammation, as well as internally curing anemia, gallstones, diuretics, hemorrhoids, and diabetic mellitus [19]. Some studies suggest that saponins, flavonoids, glucose, carboxylic acid, and other phenolic derivatives are among the biomolecules found in HA extract [20]. In this study, we aimed to produce a pectin/agar-based functional composite film containing an extract of *Hemigraphis alternata* leaf for the treatment of skin cancer wounds. Specifically, we investigated the intriguing and promising effect of incorporating HA into the AP matrix and obtained a variety of physical and functional film properties.

## 2. Materials and Methods

Agar powder was purchased from Marine Hydrocolloids Ernakulam. Citrus pectin was purchased from Sha Narendra and Sons, Chennai. The extract was prepared from *Hemigraphis alternata* leaf, which was plucked from the neighbourhood places. The remaining reagents were of analytical grade and were purchased from Kelvin Labs, India.

### 2.1. Extraction of HA Extract

For about 100 g of *Hemigraphis alternata* leaf plucked and crushed with ethanol–water mixture (1:3 ratio). The solution was then kept in a dark place for 24 h at a temperature of 4 °C. Then the extract was filtered by using a refrigerated centrifuge at 7000 rpm for 10 min. After that, the supernatant was collected and again filtered through Whatman filter paper. The obtained solution was stored at 4 °C in a dark place until it was used.

### 2.2. Preparation of APH Film

About 2 g each of agar and pectin (AP) was added in 86.3 mL of water. Then, 2 g of glycerol (30 wt% based on biopolymer) was added followed by 8.7 mL HA extract (2% wt) and the solution was homogenized gently for 10 min. The solution was heated for 30 min at 85 °C with vigorous stirring. After 30 min, the solution started gelation and was transferred into a separating funnel for casting effectively. The soluble film-forming solutions were cast on Teflon Pan and dried at room temperature for 48 h through the solution casting method.

### 2.3. Film Morphology and Texture

The surface morphology of films was characterized using a scanning electron microscope (Zeiss EVO 18) (Oxford instruments) by coating samples (3 mm × 3 mm) with a thin layer of gold and applying a 10 KV electron voltage. The morphology of membranes was observed by using Field Emission Scanning Electron Microscope (FE-SEM), Carl-Zeiss Model Ultra-microscope-55 (German). The morphological structures of AP and APH films were analyzed using SEM. The texture of the film (1 cm × 1 cm) was captured by using an optical microscope (upright optical microscope Olympus BX61) using optical stream motion software; the image was analyzed at 200 µm and 20 µm at a magnification of 15 and 500. AFM (Nanosurf easy scan2 Nanosurf AG Switzerland 23-06-154) was used to characterize the morphology of the samples more precisely. Utilizing the interactions between the tip and the sample surface, AFM is a technique capable of recreating a topographic map of the sample surface. By collecting data on the cantilever’s deflection using a laser, it is possible to obtain morphology-specific data on the sample’s surface.

### 2.4. Structural Characterization

An X-ray powder diffractometer (Bruker D8 Advance, Bremen, Germany) with Cu kα radiation (λ = 1.5405 Å) in a wide 2θ (Bragg angle) range (10 ≤ 2θ ≤ 90) was used to record X-ray diffraction (XRD) patterns of all film samples at room temperature.

### 2.5. Fourier Transforms Infrared (FTIR) Spectroscopy

The FTIR spectroscopic analysis was used to characterize the prepared films. A Fourier Transform Infrared Spectrophotometer (model: IR Affinity-1, Shimadzu, Japan) from a range of 4000–400 cm^−1^ was used for the analysis in the ATR mode in order to identify functional groups.

### 2.6. Film Color and Light Transmission

The color and transparency of the composite films with various AP concentrations on a white background plate were assessed using a colorimeter (Nix mini 2 color sensor). Three points were chosen at random to repeat the experiment after each film was cut into squares (2 × 2 cm). L* (lightness/brightness), a* (redness/greenness), and b* (yellowness/blueness) were used to describe the parameters [21]. The following formula was used to determine the total chromatic aberration:(1)$$\Delta E^{*}=\surd {\left(\Delta L^{*}\right)}^{2}+{\left(\Delta a^{*}\right)}^{2}+{\left(\Delta b^{*}\right)}^{2}$$
where Δ*L**, Δ*a** and Δ*b** are the differences between the corresponding color parameter of the samples and that of control (*L** = 70.95, *a** = 6.01, *b** = 13.96). 

Using a UV–visible spectrophotometer, the light barrier properties of AP films against visible light and ultraviolet (UV) were evaluated in accordance with the method proposed by Peng et al. [22] at a wavelength between 400 and 800 nm. The absorbance was measured at a wavelength of 600 nm using rectangular film strips in a spectrophotometer test cell. Using the following equation, the opacity of the films was determined:(2)$$O=\left(Abs_{600}\right)/L$$
while *L* is the thickness of the film specimen, *Abs* is the absorbance at 600 nm, and *O* is the opacity of the film (mm).

### 2.7. Antimicrobial Study

By adopting the disc diffusion method, an agar disc-diffusion assay was used to evaluate the antimicrobial activity in vitro against the selected pathogens, E. coli, S aureus, Enterobacter, and K pneumoniae. LB agar plates with 9 × 108 colony forming units (CFUs) of each microbe in Muller Hinton (MH) broth were covered with a film sample (7 mm in diameter) [23]. Zones of inhibition (ZOI) around each disc were measured using a digital caliper after the bacterial strains were incubated in the dark at 32.5 °C for a whole night.

### 2.8. Physical Properties of the Films

The physical properties of a polymer are in turn reliant on a wide range of additional characteristics, such as its thickness, swelling, and water contact angle. The mechanical properties of each film, including tensile strength (TS) and elongation at break percent (E), were measured with a Tinius Olsen and recorded with Horizon software. Film samples were cut into 1.5 cm × 12.5 cm strips, and the crosshead speed of the testing instrument was adjusted to 50 mm/min. The Dial Thickness Gauge 7301 Micrometer (Kawaski, Japan) was used to measure the film’s thickness with an accuracy of 0.01 mm. For each, the average of three measurements was calculated [24].

The degree of swelling was determined by slightly modifying the method proposed by Mei et al. [25]. Initially, two-by-two-centimeter samples were oven-dried at 100 degrees Celsius for twenty-four hours and then weighed (*W*1). After being immersed in 50 mL of distilled water for two hours with periodic stirring, each film sample was removed, dried on filter paper, and weighed (*W*2). Again, the films were dried to constant weight in a 70 °C air oven for 24 h and weighed. There were three measurements obtained. The degree of swelling was calculated using the formula:(3)$$\mathrm{Degree}\;\mathrm{of}\;\mathrm{swelling}=\frac{\left(W2-W1\right)}{W1}$$

The surface wettability of the film has been systematically analyzed by SEO Phoenix 300 T. Liquid contact angle analysis accurately measures the film surface’s proclivity to be wetted by liquids. The water vapor transmission rate (*WVT*) was measured gravimetrically using the standard method [18]. The composite films were cut into 6.4 cm circles and adhered to a dish filled with 50 mL deionized water, and 23 °C and 50% RH were used for the test. The weight of the dish was measured every 24 h for seven days. The water vapor transmission rate was calculated by the following equation.
(4)$$WVT=\frac{\Delta M}{\left(\Delta tA\right)}$$
where Δ*t* is the corresponding time interval, *A* is the dish mouth area, and Δ*M* is the mass change. For each samples triplicate measurement to be done.

### 2.9. In Vitro Cytotoxicity Studies

In order to assess cell viability, proliferation, and cytotoxicity, the MTT assay was used to measure cellular metabolic activity. SKMEL-26 (2500 cells/well) were seeded on 96-well plates and allowed to acclimate for 24 h to the culture conditions of 37 °C and 5% CO_2_ in the incubator. The test samples were prepared in DMEM (100 mg/mL) media and filtered with a 0.2 m Millipore syringe filter and further diluted in DMEM media, the samples were added to the wells containing the cultured cells at final concentrations of 6.25, 12.5, 25, 50, and 100 µg/mL, respectively. The untreated wells served as the control. To reduce the possibility of error, each experiment was conducted in triplicate, and the mean values were calculated. Plates were incubated for 24 h following treatment with test samples. After the incubation period, the media was aspirated from the wells and discarded. A total of 100 L of an MTT solution containing 0.5 mg/mL in PBS was added to each well. The plates were incubated for an additional two hours to facilitate the formation of formazan crystals. The supernatant was discarded, and 100 µL of 100 percent DMSO was added to each well. Using a micro plate reader, absorbance at 570 nm was measured. Two wells per plate serving as blanks were devoid of cells [26]. Each experiment was conducted in triplicate. Using the following formula, the cell viability was expressed:(5)$$Percentage\;of\;cell\;viability=\frac{Average\;absorbance\;of\;treated}{Average\;absorbance\;of\;control}\times 100$$

### 2.10. Statistical Analysis

By using SPSS 20, data were statistically analyzed in color and other physical properties of the film. The mean and standard deviation of all experimental data, which were collected from three parallel experiments (SD). For multiple comparisons, Duncan’s multiple range test and one-way analysis of variance (ANOVA) were used to see if there were statistically significant differences between the data (*p* < 0.05).

## 3. Results

### 3.1. Film Morphology

The SEM image of AP and APH are shown in Figure 1A. It is evident from the AP control film that the addition of HA extract will minimize the film’s roughness, resulting in a smoother surface. The image described here is in two different magnifications: one is in 3 KX and other one in 5 KX. The white colored particles suggest that the film containing agar and pectin is not uniformly distributed. This is because of pectin which is not easily dispersed in water. In a pure pectin film, there were minute, randomly scattered white particles on a rough, thick, and broken surface. The non-homogenized pectin debris in the pectin–glycerol matrix was recognized as the minute particles on the surface of the pure pectin film. By the addition of extract, the particles were eliminated or again, their size was reduced through ultrasound sonication method and uniformity of APH increased. The morphology of the films was examined by the FE-SEM images, which are displayed in Figure 1B. A rough, uneven surface morphology is observed, with few aggregates scattered on the surface of the control film. It is evident that the surface of APH contains more moisture than AP; this is because of the incorporation of extract. This also results in increasing the hydrophilicity of bio polymer. 

For texture analysis of the film, an optical microscope was used. Figure 1C shows the optical images of representative films. Optical images of AP and APH were observed under polarized light. The addition of extract causes a significant change in the base matrix. As suggested by the SEM and FE-SEM images, the addition of extract enhanced the uniformity of the film. The optical image reveals that the control film is composed of particles. The particle size decreased significantly due to the addition of extract. The use of extract makes the entire biofilm smooth, hence enhancing its overall nature. Figure 1D depicts the AFM image of both AP and APH film samples, and it is evident that the APH film has some minor surface roughness, but not as much as the AP contact surface of the blended film. The fact that both modified surfaces were exposed to air while being modified suggests that the original surface characteristics are not significant during modification. The AFM image demonstrates that the AP modified blended film (APH) was smoother than the unmodified film. 

### 3.2. Structural Characterization

The X-ray diffraction patterns of the films with HA extract contents were examined and are shown in Figure 2, in order to analyze the change in the structure of AP-based films. All diffractograms exhibited a broad diffuse band centered at approximately 2θ = 22° and a low-intensity peak at 2θ = 15°. When HA was added, the peak at 14° had some small changes and the main broad peak at 2θ = 20° slightly decreased in intensity. The addition of extract to the AP matrix did not significantly alter the APH composite. In the current study, both films exhibited a broad peak at 15° confirming the presence of some crystalline structure due to the addition of pectin. With the exception of a broad peak at 19 and a weak shoulder at about 14, there were no sharp peaks in agar, indicating that it is predominantly amorphous [27]. The minor crystallinity of agar is due to the double-helical conformation formed by intermolecular hydrogen bonds [28]. Inclusion of HA into the AP film did not significantly alter the peak positions in the XRD pattern.

### 3.3. Fourier Transforms Infrared (FTIR) Spectroscopy

The interaction between agar and pectin with *Hemigraphis alternata* is shown in Figure 2. A broad band obtained in the range from 3000 to 3500 cm^−1^ is OH and CH stretching. The agar has an extremely low methylation degree, as evidenced by the band at 3313 cm^−1^ representing OH groups, the absorbance at 2944 cm^−1^ representing CH_2_ groups, and the lack of significant absorbance at 2845 cm^−1^ (O-CH_3_) [29]. The C=O component of a pectin ester bond was responsible for the peak at 1743 cm^−1^. The symmetric and asymmetric stretching of the carboxyl groups in the pectin structure were attributed to two distinct bands at 1638 and 1422 cm^−1^, respectively [30]. The peaks at 1034 and 934 cm^−1^ indicate C-O stretching residual carbon of 3,6-anhydro-galactose. And the peak 853 cm^−1^ related to the C-H stretching of β-galactose [31]. The addition of extract does not affect the control film by much, only some slight difference in some peaks is observed. One of the variations that may be monitored with AP film is the peak at 1231 cm^−1^ indicating C-O-C stretching in HA, and one characteristic carbonyl stretching (C=O) band is observed at 1640 cm^−1^. 

### 3.4. Film Color and Light Transmission

The edible film’s color and transparency serve as two important gauges of both its general appearance and consumer approval. Table 1 lists the impact of HA addition on the color coordinates (L*, a*, and b*), total color difference (E), and films’ opacity (O). Extract incorporation in edible films made them appear more transparent and clearer. The color and transparency of biofilms were only slightly changed by the addition of HA extract. The AP and APH only slightly differed in value at 600 absorbance, and the value dropped with the addition of extract. The neat pectin/agar was transparent, and the film was slightly yellow by the addition of extract. The appearance of the leaf extract is brown in color. Incorporating extract to AP film changes the appearance of control film. 

Table 1 displays the color values for the films made with agar/pectin. According to Roy et al., the addition of GSE did not alter the film’s brightness (Hunter L value), but the addition of alizarin resulted in a significant reduction in brightness because of the deep yellow hue of alizarin [32]. In this instance, the addition of HA extract changed all values and produced a deep yellow color, this confirming the increase in value of b. The parameter b indicates the yellowness/blueness in the film. Comparing with the control film, the value increased up to 35.06 ± 0.64. The addition of extract decreased the L value while raising the a and b values. The control film had better color values but less transparency. Increased extract concentration results in greater UV blocking properties. However, compared to the control film, the composite films’ overall transparency was better.

### 3.5. Antimicrobial Activity of the Synthesized Wound Healing Material

Antimicrobial activity of AP and APH was obtained by disk diffusion method with four different foodborne pathogenic bacteria: *E. coli*, *S. aureus*, *Enterobacter* sp. and *K. pneumoniae*, where *E. coli* and *Enterobacter* sp. and *K. pneumoniae* are Gram-negative and *S. aureus* is Gram-positive organism. Agar and pectin film has no effect with any of the organisms. There is no antimicrobial effect with AP and APH in *E. coli* and *S. aureus*. The APH film shows a small zone of inhibition over Enterobacter (10 mm) and *K. pneumoniae* (8 mm). Due to the addition of extract, the antimicrobial property increased. By this concentration of extract (2%) in AP, severe antibacterial property to film is shown. Anitha et al. reported that an ethanolic extract from the leaves of *Hemigraphis alternata* exhibited antibacterial activity against the bacterial pathogens *K. pneumonia*, *S. typhi*, *B. cereus*, *S. marsecens*, *S. pyogenes*, *Acinetobacter* sp., and *Enterobacter* sp. [33].

### 3.6. Physical Properties of the Films

Some physical properties such as thickness, swelling behavior, and WVP are shown in Table 2. There is only a slight difference in thickness by adding extract into the AP matrix. The thickness of the film is attributed by the addition of chemical components in the matrix. Incorporating red cabbage extract increased the thickness of corn starch, according to Prietto et al. [34]. From Table 2, the thickness of the film is found to be 0.17 and 0.19 mm. Here, the value indicates the addition of HA extract, which reduces the thickness as compared to the control. Norajit et al. reported that the addition of ginseng extract to alginate-based films significantly increased the thickness of the films [35]. Increased tensile strength and elongation are attributable to thicker walls; adding HA may have thickened the linked walls, which increased APH’s strength and elongation. Compared to control films, the amount of solids in the film-forming emulsion influences the thickness of the film [36].

The swelling ratio (SR) represents the water holding capacity of the film. From the above table, the value of SR of APH decreased by the addition of HA extract. In et al. reported that, due to the incorporation of basil leaf extract, the swelling index of the film decreased. A decrease in the swelling index may be attributable to induced interactions between the phenolic and AP matrices, thereby decreasing the swelling index [37].

In clinical settings, the prediction of the water exchange for a wound covering is made easier by the WVP of a wound covering, which is a more precise estimate of permeability than the WVTR is. Water vapor permeability indicates the relativity of high water vapor barrier property of AP and APH films. When compared to APH, WVP of AP has a higher water vapor barrier property. The mean values of WVP for the AP film with value corresponds to 0.46 ± 0.2 × 10^−9^ gm/m^2^·Pa·s and APH 0.49 ± 0.1 × 10^−9^ g·m/m^2^·Pa·s. The improvement in the composite film’s water vapor barrier properties or decrease in WVP are most likely due to the formation of a dense and compact network between the filler and polymer matrix and an increase in hydrophobicity [32]. The fabricated material in this instance has a lower water vapor permeability. The swelling study and water contact angle measurement show that APH have a higher water absorption capacity. As a result, it is ideal for treating wounds.

An essential characteristic of biomaterials is surface wettability, which has an impact on how well cells adhere to surfaces as well as how they grow, migrate, and remain viable. The surface tendency of water droplets is determined using the water contact angle. Using this technique, the angle between the film surface and the water droplet is calculated. This characterization revealed the film’s hydrophobicity or hydrophilicity and wettability. We used contact angle (θ) to measure where the liquid/vapor interface meets the surface of the film. To measure, deionized water was added to a platform with a piece of film at room temperature. A micro-syringe produced the droplets (1–2 µL) on the surface of film. All of the characteristics of the biomaterial, such as water absorption, cell interactions, and in vitro and in vivo degradation, are necessary for wettability. Where the water contact angle falls below 90°, the solid surface becomes hydrophilic, and where it rises above 90°, the solid surface is termed hydrophobic [38]. The obtained contact angle images are shown in Figure 3. The values revealed that the film having good water absorption capability Table 2. The addition of extract increases the hydrophilicity of control film. As previously reported, the contact angle for agar and pectin blended films is hydrophilic in nature [39]. On the basis of this characteristic, the film is effective as a wound healing material. 

Figure 4 shows that when strain increases, stress rises until it reaches a maximum value before falling. The findings showed that adding the extract improves the mechanical characteristics. According to Roy et al., the inclusion of GSE and MNP enhanced the polymer’s strength [32]. Consequently, HA extract improves the mechanical characteristics of APH wound healing materials.

### 3.7. Anticancer Activity of Films

The MTT assay measures cellular metabolic activity as an indicator of cell viability, proliferation, and cytotoxicity for the purpose of evaluating the anticancer activity of biofilm. SKMEL-26 cancer cells administered varying concentrations of the sample APH, exhibiting a dose-dependent reduction in the cell viability. The results show that, by the increase in concentration, the cells were reduced at a concentration of 100, corresponding to 38.01%. The IC50 value indicates that the film is biodegradable and effective at killing melanoma cancer cells. The IC50 value is the half maximal inhibitory concentration of the sample. The average absorbance of the various concentrations (6.25–100 μg/mL) of the test sample from Figure 5 was plotted to obtain the equation for slope (y = mx + C), which was then used to calculate IC50 values. The IC50 value was obtained as 68.28 µg/mL of the sample (Table 3). Therefore, the usage of this scaffold in biomedical sector will increase the demand of films. Figure 5 shows the images captured by an MTT optical microscope of the growth of SKMEL-26 cancer cells after a 48-h culture period in the presence of different concentrations of the film samples.

## 4. Conclusions

As a therapeutic drug, HA extract was incorporated into the agar/pectin-based therapeutic functional films. We have developed a biopolymer wound healing material with properties such as biodegradability, antibacterial, water absorption capacity, and nontoxicity in response to the prevalence of synthetic wound healing materials. The entire preparation procedure is based on eco-friendly principles. This signifies an increasing demand for natural polymer wound dressings in the medical field. The use of agar/pectin-blended functional films for packaging was mentioned in prior research. The cytotoxicity analysis demonstrated the film’s anticancer properties and the necessity for additional in vivo research. To improve the functionality of the film, the extract with therapeutic characteristics was added. The inclusion of extract enhances the film’s hydrophilicity. It absorbs water and acts as a barrier against the outer atmosphere as a result of this behavior. According to our analysis of this study, the addition of a concentrated extract improves the results of antimicrobial studies. Additionally, further research is required for the development of intelligent wound care materials.

## Figures and Tables

**Figure 1 polymers-15-00115-f001:**
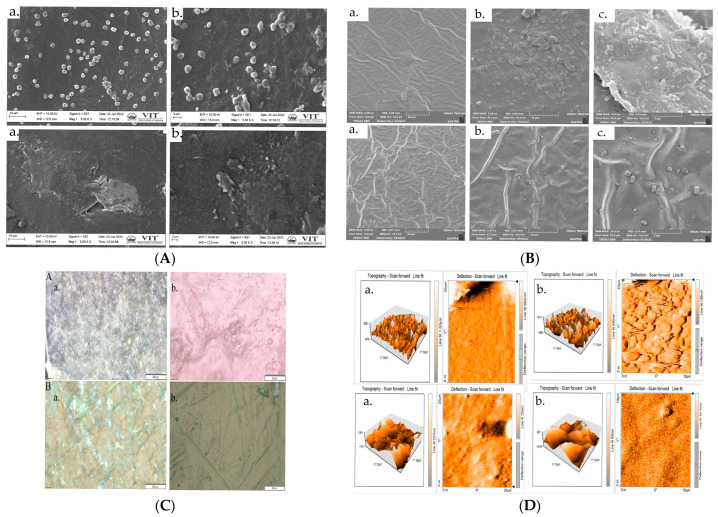
(**A**) SEM image of AP and APH at (a) 10 μm and (b) 2 μm. (**B**) FESEM image of AP and APH at (a) 20 μm (b) 10 μm and (c) 5 μm. (**C**) Optical image of AP and APH at (a) 200 μm and (b) 2 μm. (**D**) AFM image of AP and APH at (a) 10 μm and (b) 25 μm.

**Figure 2 polymers-15-00115-f002:**
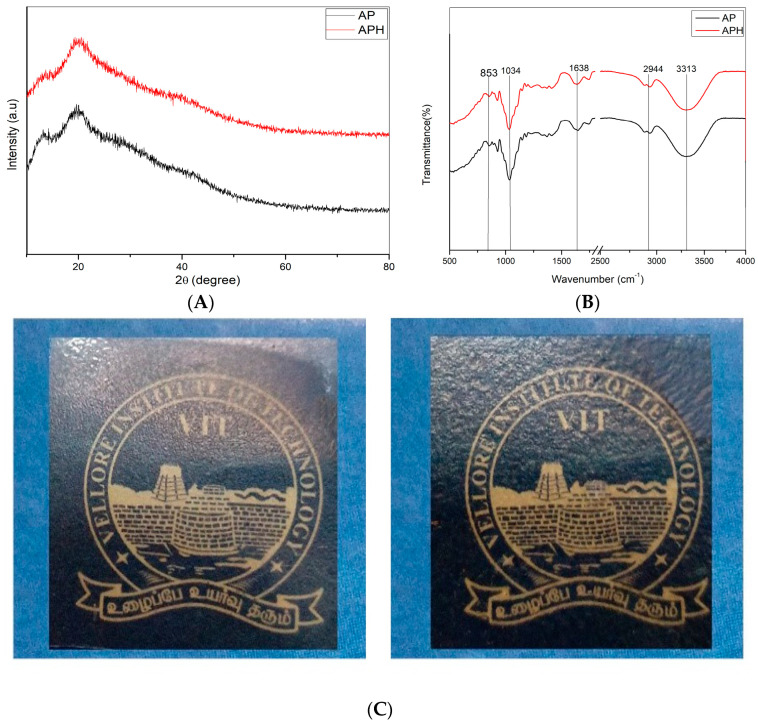
(**A**) XRD image. (**B**) FTIR spectrum. (**C**) Film transparency of AP and APH films.

**Figure 3 polymers-15-00115-f003:**
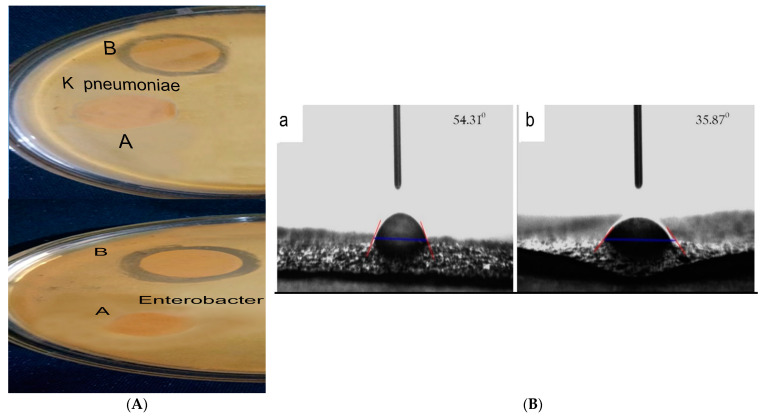
(**A**) Antimicrobial property of film’s with different bacterial culture A) AP and B) APH. (**B**) Water Contact Angle of (a) AP and (b) APH.

**Figure 4 polymers-15-00115-f004:**
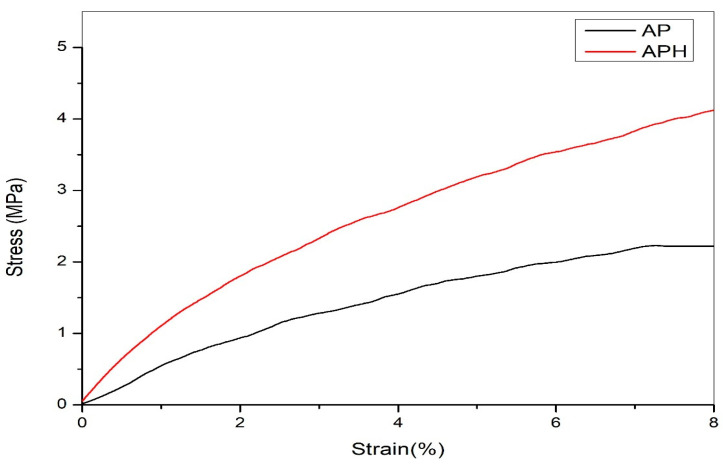
Stress Strain graph of AP and APH film.

**Figure 5 polymers-15-00115-f005:**
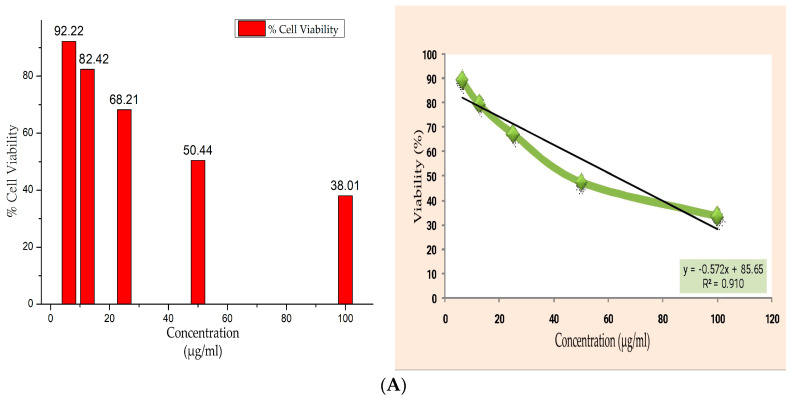
(**A**) Cell viability % of samples with different concentration and line graph of viability % with concentration by using equation y = mx + c. (**B**) MTT optical microscopy images of SKMEL-26 cancer cell growth after 48 h culturing in presence of the film. Cell viability % with concentrations of 6.25, 12.5, 25, 50, and 100 µg/mL.

**Table 1 polymers-15-00115-t001:** Film color parameters and opacity.

Film	L	a	b	ΔE	Opacity (A mm^−1^)
AP	70.35 ± 0.94 ^c^	5.74 ± 0.15 ^a^	13.85 ± 0.16 ^a^	1.13 ± 0.85 ^a^	3.44
APH	63.27 ± 0.65 ^b^	11.03 ± 0.89 ^c^	35.06 ± 0.64 ^d^	23.03 ± 0.55 ^c^	3.11

^a^ Standard deviation divided by the mean is used to illustrate the values. There is no statistically significant difference between any two means that are followed by the same letter in the same column (*p* > 0.05).

**Table 2 polymers-15-00115-t002:** Thickness, WCA, WVP, and swelling ratio of AP and APH films.

Film	Thickness (mm)	WCA (θ)	WVP (×10^−9^ g·m/m^2^·Pa·s)	Swelling Degree (%)	Tensile Strength (MPa)	Elongation of Break (%)
AP	0.17 ± 0.01 ^a^	54.31°	0.46 ± 0.2 ^a^	1.408 ± 0.46 ^a^	2.22 ± 0.2 ^a^	9.70± 2.1 ^ab^
APH	0.19 ± 0.01 ^b^	35.89°	0.49 ± 0.1 ^a^	1.786 ± 0.10 ^b^	4.32 ± 0.3 ^a^	10.9± 1.5 ^ab^

**Table 3 polymers-15-00115-t003:** Concentration and percentage of viability of APH film and its IC50 value.

Concentration (µg/mL)	Percentage of Viability
6.25	92.22 ± 0.673
12.5	82.42 ± 0.555
25	68.21 ± 0.459
50	50.44 ± 0.339
100	38.01 ± 0.256
IC50	68.28

## Data Availability

Not applicable.

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
