# Peer review of "Hemigraphis alternata Leaf Extract Incorporated Agar/Pectin-Based Bio-Engineered Wound Dressing Materials for Effective Skin Cancer Wound Care Therapy"

_polymers, 2022, doi:10.3390/polym15010115_

Round 1
Reviewer 1 Report
In the review of the manuscript titled Hemigraphis Alternata Leaf Extract Incorporated Agar/ Pectin Based Bio-engineered Wound Dressing Materials for Effective Skin Cancer Wound Care Therapy. The authors have provided a good description and the methodology is also fine. I would like to see this article publish but after some questions as follow;
1. Why the film containing agar and pectin showed non-uniform distribution?
2. Why did the particle size decrease significantly due to the addition of extract?
3. How did the authors determine the surface tendency of water droplets?
4. Why did the findings show that adding the extract improves the mechanical characteristics?
Reviewer 2 Report
The paper is well written and well crafted. However, it needs a little reconstruction. There are a lot of typos/errors in the manuscript.
Please make the graphs uniform in terms of x and y axis, font size, font style etc.
Line 485, Why the reference 41 is mentioned but it is not provided.
Line 344, the standard deviation of percentage viability of cells is not provided.
Line 344, The y-axis of cytotoxicity detection MTT assay is not clear.
Use updated references of the last five years should be added.
